# High-Strength, Waterproof, Corrosion-Resistant Nano-Silica Carbon Nanotube Cementitious Composites

**DOI:** 10.3390/ma13173737

**Published:** 2020-08-24

**Authors:** Hao Li, Yongmin Shi

**Affiliations:** 1Shaanxi Key Laboratory of Chemical Additives for Industry, College of Chemistry and Chemical Engineering, Shaanxi University of Science and Technology, Xi’an 710021, China; wheat-field@sust.edu.cn; 2School of Earth and Space Sciences, Peking University, No. 5 Yiheyuan Road, Haidian District, Beijing 100871, China

**Keywords:** silane coupling agent, nano-silica, carbon nanotube, composite structure, cementitious composite

## Abstract

This study aims to prepare a nano-silica-carbon nanotube (NS-CNT) elastic composite using NS (nano-silica), CNTs (carbon nanotube), and (D_3_F) trifluoropropyltrimethoxysilane. The results show that the activated NS could promote the hydrolysis of D_3_F. Polymerization products of nano-silica and D_3_F are uniformly adhered onto the surfaces of CNTs, thereby forming a NS-CNT composite. The composite is composed of irregular ellipsoids of 3–12 μm in length and 2–10 μm in diameter. The activated NS-CNT composite material effectively promotes the further hydration of (CaOH)_2_ in the cement to form hydrated calcium silicate, and further dehydration–condensation between the surface hydroxyl group of the composite material and the inherent hydroxyl group of (CaOH)_2_. The cementitious composite-based composites containing the activated NS-CNT exhibit high mechanical strengths, high water resistances, and good durability and corrosion resistance. The chemical characterizations reveal the morphology, nucleation mode of the composite, and its influence on the hydration structure and products of cementitious composite.

## 1. Introduction

Attributed to its excellent structural properties, cementitious composite is widely used as a building material, such as in the construction of bridges, tunnels, dams, railways, and wharfs. At the extraction of oil, Cl^−^ can degrade the rebar performance and shorten the service life of the reinforced cementitious composite [1]. Hence, to enhance the properties of cementitious composite, nanomaterials in the size range of 1–100 nm have been added in various commercial applications [2]. Due to their large surface areas, nanomaterials can impart beneficial properties to the cementitious composite paste, such as a good early bond strength, reduced permeability, accelerated cementitious composite hydration, and controlled fluid loss. As such, more calcium silicate hydrate (CSH) and less calcium hydroxide are produced in the presence of the nanomaterials, which makes the cementitious composite denser [3].

Nanomaterials possess small volume and adding nanomaterials such as nano-silica (NS), Sardon et al. used (3-aminopropyl) triethoxysilane (APTES)-modified SiO_2_ to reduce the agglomeration of NS significantly [4,5,6,7,8,9,10,11,12,13]. The addition of nano-titanium dioxide (nano-TiO_2_) [14], nano-iron (nano-Fe_2_O_3_) [15], nano-alumina (nano-Al_2_O_3_) [16], carbon nanotube (CNTs) [17,18,19], and nano-clay particles [20] into cementitious composite can improve its overall properties. Among these nanomaterials, NS and CNTs are particularly significant in cementitious composite [21,22]. NS is composed of very fine SiO_2_ particles that are nearly 1000 times smaller than ordinary cementitious composite particles, with high pozzolanic activity. Moreover, the addition of NS curing paste can shorten the setting time. Therefore, NS is widely used to increase the impermeabilities of pastes and the mechanical properties of hardened materials. On the other hand, CNTs have garnered increasing attention due to their relatively low densities, which can improve the thermal properties of the cementitious composite. Their positive effects on the performance of cementitious composite-based composites are mainly affected by two phenomena: hydrate nucleation and filling effects. However, the greatest challenges in implementing these nanomaterials in cementitious composite are the compatibility and dispersibility of these nanomaterials in the cementitious composite, which warrants an urgent resolution.

Physical methods, such as mechanical force and ultrasonic dispersion, and chemical methods are mostly used in the preparation of composite. For the chemical methods, modifiers are added to reduce the hydroxyl groups on the surface of NS through chemical reaction, and to reduce the nanoparticles’ agglomeration. For instance, Mahadik et al. studied the preparation of TEOS-based silica gels by a two-step sol–gel method followed by surface-treating the fumed SiO_2_, and modification of its surface with a silylating agent. Hydrophobic aerogel comprising trimethylchlorosilane (TMCS) and hexamethyldisilazane (HMDZ) can be realized using an ambient pressure drying method [23]. As such, these functionalized NS and CNTs can chemically interact with the cementitious composite particles, which in turn can positively influence the hydration process. However, chemical modification of CNTs may reduce their mechanical strengths, which could reduce the overall properties of the composites. This is because the chemicals used in the functionalization of CNTs can have a negative impact on the hydration kinetics. Studies by Yakovlev and Mendoza-Reales et al. have shown that the combination of CNTs and NS can accelerate the hydration kinetics, resulting in greater production of C-S-H [24]. Due to the chemical compatibility of the NS and the C-S-H matrix, SiO_2_ located on the surfaces of the CNTs can enter the matrix and form aggregates. Therefore, due to the improved bonding between the CNTs and NS in the matrix, enhanced load transfer capabilities can be achieved. Monfared et al. have shown that the incorporation of nanoparticles enhanced the fiber–matrix interfacial strength, toughened the surrounding matrix, and improved resin adhesion to the fiber, which increased the tensile properties of incorporated composites with hybrid nano-fillers [25,26,27,28,29,30]. Rui Wang and others proved that the incorporation of the as-prepared nanosilica/graphene oxide (m-SGO) hybrid into epoxy resin (EP) resin not only obviously increases the flame retardancy, mechanical, and thermal stability properties, but also endows EP resin with high thermal conductivity, low dielectric loss, and high dielectric constant. This study yielded similar experimental results [31]. MR Ayatollahi also showed that the combination of carbon nanotubes and carbon nanotubes strengthens the fiber–matrix interface strength, and the stiffness of the multi-scale composite material [32]. Miaomiao Hu discovered through research that the nanosilica particles on MWCNT-81 G/VMQ(GO) surface formed a physical barrier to maintain good dispersion of GO and NS. This is similar to the findings of this experiment [33]. Bratati Pradhan also improved the performance of MWCNT-G/VMQ at the same time, because Nanocomposites can be attributed to the synergistic effect of hybrid fillers in pore solution, respectively [34]. Majid R Ayatollahi also studied the effect of carbon nanotubes and nanosilica on the tribological properties of carbon fiber cloth composite materials and concluded that which indicates enhan cementitious composite in bonding strength between carbon fiber and epoxy matrix due to the interfacial reinforcing action of the nano-particles [35].

In this study, the effect of NS and CNT composite structure on the properties of cement materials was systematically studied, including the mechanical properties, water resistance and corrosion resistance of the cement base. Nuclear Magnetic Resonance (NMR) (Bruker, Switzerland, Germany) and Fourier Transform In-fared (FTIR) (Nicolet 5700. Bruker, Switzerland, Germany) were used to study the dehydration condensation of nano-silica and trifluoropropyltrimethylsiloxane. Scanning electron microscopy (SEM)( FEI Verios 460, Hillsboro, OR, USA) and X-ray Diffraction (XRD)(Bruker D8-Discover instrument, Karlsruhe, Germany), X-ray Photoelectron Spectroscopy (XPS)(British AXIS SUPRA), TG equipment, and calorimetry have been used to study the influence of nano-silica and CNT composite structure to enhance the microstructure and hydration of the cement slurry. The freeze–thaw resistance experiment was used to study the influence of NS and CNT composite structure on the corrosion resistance of cement materials. The contact angle experiment was used to study the waterproofness of cement materials. This article also introduces the experimental procedures, results analysis, and discussion in detail.

## 2. Materials and Methods

### 2.1. Materials

NS and CNTs were purchased from Beijing DK Nano Technology Co., Ltd. (DeKeDaoJin Nanomaterials Ltd., Beijing, China)., and their relevant parameters are shown in Table 1.

For experimental groups 1-1 to 4-4: NS was added to trifluoropropyltrimethoxysilane, which was heated to 80 °C and stirred for 2 h. After which, CNTs were added, followed by continuous stirring at 80 °C for 2 h. Sixteen experimental groups were configured according to the mixing ratios shown in Table 3 (graphitized CNTs (G-CNTs) were added to experiment group 1; hydroxylated graphitized CNTs (OH-G-CNTs) were added to experimental group 2; short arm CNTs (S-MWNTs) were added to experimental group 3; hydroxylated short arm CNTs (OH-S-MWNTs) were added to experimental group 4. Control group 5 was pure cementitious composite without any additives. For control group 6, NS was added to trifluoropropyltrimethoxysilane and continuously stirred for 2 h at room temperature. CNTs (OH-G-CNTs) were then added with continuous stirring for 2 h at room temperature.

The mixed cementitious composite paste was added to the JSF 550 agitator (Kunshan Shunnuo Instrument Co., Ltd.) (Kunshan, China) at a water–cementitious composite ratio of 0.38, with a rotation speed of 800 r/min and a mixing duration of 20 min. After mixing, the paste was removed and placed into the molds. Mechanical properties were tested with a cementitious composite triplex mold with dimensions of 40 mm × 40 mm × 160 mm. The freeze–thaw experiment used a 20 mm × 20 mm × 20 mm mold. The electrochemical corrosion experiment used a mold with a diameter of 50 mm and a height of 30 mm, with a cleaned iron bar with a diameter of 10 mm and a length of 80 mm inserted. All samples were kept in a temperature and humidity chamber with a temperature of 25 °C and a humidity of 98% for 28 days.

### 2.2. Analytical Methods

#### 2.2.1. Materials Characterization Techniques

The AVANCE III HD 400 Bruker spectrometer (Switzerland Bruker, Karlsruhe Germany) was used for the 1H nuclear magnetic resonance (NMR) test, using deuterated chloroform (ClD3) as the solvent. Fourier Transform In-fared (FTIR) was recorded using a FTIR spectrometer (Nicolet 5700, Bruker, Switzerland, Germany), between wavenumbers of 400 and 4000 cm^−1^ with a resolution of 0.1 cm^−1^. X-ray Photoelectron Spectroscopy (XPS) was conducted using a British AXIS SUPRA instrument for the qualitative analysis of the surface elements and their valence states. The crystalline phases were analyzed with X-ray Diffraction (XRD) spectroscopy (Bruker D8-Discover instrument, Germany), in the 2θ range of 10–70° with a step size of 0.02° and a scan speed of 0.25 s/step. The thermal behaviors of the cementitious composite paste and nano-silica before and after treatment were analyzed using a Netzsch TG209 F3 TG equipment(Beijing, China), conducted at a heating rate of 10 K/min to 600 °C in a platinum crucible under an ambient environment of 100 cm^3^ of N_2_. A TAM Air isothermal calorimeter was used to assess the evolution of the heat flow of the cementitious composite pastes. Immediately after mixing, around 10 g of cementitious composite paste was placed into standard glass containers, and the paste was loaded into the isothermal calorimeter. The cementitious composite hydration process was stopped via freeze drying after 28 days. A quantity of 10 mg of the powder sample was first subjected to gold-sputtering, before observation under Scanning Electron Microscopy (SEM) using the FEI Verios 460. Brunauer–Emmett–Teller (BET) analysis was carried out using a Micromeritics 2460(Switzerland Bruker, Germany). Flexural and compressive tests were carried out using a TYE-300D cementitious composite mortar flexural and compressive test machine (Wuxi Jianyi Instrument Machinery Co., Ltd.) (Zhejiang, China). The corrosion resistance test adopts a self-designed device, which contains two electrodes and a power plug. The two electrodes are, respectively, connected to the copper rod in the cement block sample, and the other end is connected to the copper rod and placed in salt water, and 220 V AC is connected. The corrosion resistance experiments were carried out in NaCl (3.5 wt%) at 26 V for 20 s. The contact angle measurement used the OCA series video optical contact angle measuring instrument developed and produced by the German Dataphysi company. Contact angle measurement range: 0–180°; measurement accuracy: ±0.1°.

#### 2.2.2. Anti-Freeze–Thaw Damage, Electrochemical Corrosion Test and Contact Angle Test

Freeze–thaw damage was tested in a cementitious composite freeze–thaw machine (KDS-28, Suzhou Donghua Test Equipment Co.) (Zhejiang, China), according to the China National Standard GB/T50082-2009. Before the accelerated freeze–thaw process, cubic cementitious composite samples of 20 × 20 × 20 mm^3^ were immersed in water at 18–22 °C for 2 days. Natural water absorption was then measured according to the following method: The sample was first dried at 110 °C for 24 h, and the mass of the dried sample (m_0_) was measured. The dried sample was later immersed into water for 48 h. The sample was then collected and padded dried. The mass of the immersed sample (m_s_) was measured immediately. The natural water absorption was calculated as W = (m_s_ − m_0_)/m_0_ × 100%. The method to determine the anti-freeze–thaw damage was as follows: The sample was inserted into the instrument at an initial temperature of 6 °C. The temperature was later decreased to −15 °C after 1 h. After which, the temperature was increased to 6 °C in the next hour. This process constituted one cycle. After every 24 cycles, the sample was removed, and its mass was measured after the surface residue was peeled off. In the first, third, fifth, and seventh sets of 24 cycles, the instrument was turned off when the temperature decreased to −15 °C, and the masses measured were denoted as m_1_, m_3_, m_5_, and m_7_, respectively. In the second, fourth, and sixth sets of 24 cycles, the instrument was turned off when the temperature increased to 6 °C, and the masses measured were denoted as m_2_, m_4_, and m_6_, respectively. The mass loss was calculated as follows: W = (m_n_ − m_n−1_)/m_n_ × 100%, where n is the set number. All the measurements were accurate to 0.01 g. To study the anti-corrosion ability, the carbon steel bar coated with cementitious composite was electrochemically corroded at 26 V for 20 s in a 3.5 wt% aqueous NaCl solution. The cementitious composite was subsequently broken to observe the macro-morphology of the carbon steel bar. Before the test, all the samples were immersed in 3.5 wt% aqueous NaCl solutions for 24 h. For the contact angle measurements, the DSA100 instrument and pendent drop method were used to measure the dynamic contact angle of the cementitious composite-based composites.

## 3. Results and Discussion

### 3.1. Mechanical Properties Analysis

Flexural and compressive tests were carried out using a TYE-300D cementitious composite mortar flexural and compressive test machine (Wuxi Jianyi Instrument Machinery Co., Ltd.) (Zhejiang, China). The dimensions of the specimens used in the test were 40 mm × 40 mm × 160 mm, and they were made of cementitious composite paste or mortar, according to the China National Standard JTG E30-2005, The Methods of Cementitious Composite and Cementitious Composite for Highway Engineering. As shown in Figure 1, the measured compressive and tensile strengths are 102.04% and 140.47% higher than those of the control group, respectively, and 53.51% and 71.43% higher than those of the group without hydrothermal activation treatment. Since the composite structure of nano-silica-carbon nano-tubes is successfully embedded in the voids of cement materials, its mechanical properties have been greatly improved.

### 3.2. NMR, FTIR, and XPS Analysis

Figure 2a shows the ^1^H NMR spectra of the specific molecular formula of trifluoropropyltrimethoxysilane. To prevent hydrolysis and to determine the original molecular formula of trifluoropropyltrimethoxysilane, deuterium chloroform was used as the solvent. H standard peaks are observed at 1H (δ = 3.52 ppm, d, *J* = 1.2 Hz), 2H (δ = 2.12–1.99 ppm,), and 3H (δ = 0.81–0.72 ppm, m). Figure 2b shows the ^1^H NMR spectrum of trifluoropropyltrimethoxysilane without activated NS. The -O-CH_3_ group as 1H is clearly observed at (δ = 3.78–3.41 ppm, m), but peaks at 2H (δ = 2.25–2.05 ppm, m) and 3H (δ = 0.99–0.78 ppm, m) are also detected. The splitting in the spectrum could be caused by the sample concentration, which is consistent with the experimental results reported by Gabrielli et al. [24,25]. Figure 2c shows the ^1^H NMR spectrum of trifluoropropyltrimethoxysilane containing activated NS. The peaks of the -O-CH_3_ group at ^1^H (δ = 3.46 ppm, s), 2H (δ = 2.11, q, *J* = 9.2, 8.1 Hz), and 3H (delta = 0.95–0.74 ppm, m) are observed, and peaks of -OH at (δ = 4.99–4.69 ppm, m, OH) due to the hydrolysis of -O-CH_3_ are also observed, indicating that the activated NS can accelerate the hydrolysis of trifluoropropyltrimethoxysilane.

To characterize the effects of NS on the hydrolysis and condensation of trifluoropropyltrimethoxysilane, as well as the effects of NS-CNT structure on the cementitious composite hydration, the FTIR spectra of G-CNT10.8 + NS7.2, OH-G-CNT10.8 + NS7.2, S-WMNT10.8 + NS7.2, OH-S-WMNT10.8 + NS7.2 samples, and control groups 5 and 6 are analyzed (Figure 2d). The results show that the cementitious composite composites with activated NS-CNT contain free hydroxyl groups, which do not form hydrogen bonds at ca. 3600 cm^−1^. When hydrogen bonds are formed, the absorption peaks shift to the low frequency range of 3300–3500 cm^−1^, forming a strong and broad OH stretching vibration absorption band. This further indicates that the activated NS can promote the hydrolysis of trifluoropropyltrimethoxysilane. Due to the hydrogen bonding, dehydration, and condensation of Si-OH after hydrolysis with the -OH group on the surface of NS are achieved. This phenomenon is further confirmed by the increase in the number of Si-O-Si peaks between 1000 and 1200 cm^−1^. However, free hydroxyl groups are not observed in the control group 5, and only a small amount of free hydroxyl groups are observed in control group 6. The obvious peak area of the hydroxyl groups in control group 6 is larger than those in the control group 5 and the experimental groups. The peaks in the 3300–3500 cm^−1^ range are attributed to the -OH groups in calcium hydroxide [11]. The peak at 1630 cm^−1^ is ascribed to the stretching vibration of the C = C, which corresponds the CNTs. Due to the uniform adhesion of the NS, corresponding SP^3^ hybridization occurs on the surfaces of the CNTs, and the number of defects increases. This is indicated by the peaks at 2923 and 2853 cm^−1^ in the hydrothermally activated NS-CNT composite, which correspond to the stretching vibrations of C–H in –CH and –CH_2_. These peaks are not observed in control group 5, while weak peaks are observed in control group 6. This indicates that only a small amount of NS are adhered onto the surface of the CNTs, resulting in weak defects.

XPS is used to verify the surface elements of the cementitious composite materials. Figure 3e shows that C, O, Ca, and Si are present in the cementitious composite. As shown in Figure 3a, the Si 2p XPS spectrum reveals a main peak located at 102.3 eV, which indicates the existence of Si-O [26]. The C 1s XPS spectrum shows peaks at 286.2, 288.7, 293.2, and 296.0 eV, which can be assigned to -CF_2_ and -CF_3_ (Figure 3b), respectively. This proves that the particles on the cementitious composite are successfully coated with trifluoropropyltrimethoxysilane.

The Ca 2p XPS spectrum, presented in Figure 3c, reveals the Ca 2p_3/2_ and Ca 2p_1/2_ peaks at 347.0 and 351.1 eV, respectively. This indicates the existence of the hydration product, i.e., CaOH, and hydrated calcium silicate [26]. The C1s XPS spectrum shows a peak at 284.6 eV, which can be assigned to CC/CH/C–Si. Most carbon atoms are sp^2^ hybridized, while the C–C peaks are the main peaks, which correspond to the graphene phase. This proves that CNTs are successfully coated onto the surfaces of the cementitious composite particles [25]. As shown in Figure 3d, the O 1s XPS spectrum shows a main peak at 532.6 eV, which suggests the presence of Si-O and Si-O-Si bonds in the cementitious composite [27].

### 3.3. XRD, TG, and Isothermal Calorimetry Analysis

The phase compositions of the cementitious composite were investigated with XRD, and the results are shown in Figure 4a–d. The characteristic peaks of the hydrated cementitious composite include various main products such as calcium hydroxide, calcium silicate hydrate, tricalcium silicate, dicalcium silicate, and tricalcium aluminate. This result proves that NS-CNT does not promote the hydration of cementitious composite to form new crystalline phases. Similar characteristic peaks are shown in Figure 4e, but it is evident that the calcium hydroxide peak at 2θ = 18° disappears after the addition of NS-CNT. In addition, compared with the characteristic peaks of the composites without hydrothermal treatment, the characteristic peaks of tricalcium silicate, dicalcium silicate, and tricalcium aluminate significantly increase in size, even though the peaks of calcium hydroxide disappear after hydrothermal treatment. These results confirm that the NS-CNT composite after hydrothermal treatment could further accelerate the hydration of the cementitious composite hydration product, i.e., calcium hydroxide, to form hydrated calcium silicate due to its own nucleation effect. Comparing the XRD patterns of the four composites with different types of CNTs, NS-CNT composite can be formed so as to promote the further hydrolysis of calcium hydroxide. No observable difference in all XRD patterns suggests that the crystalline phase of the cementitious composite is not affected by the type of CNT added. The best mixing ratio of CNTs to nano-silica is 10.8:7.2, as shown by the compressive and tensile strengths [28].

The effects of the NS-CNT composite on the thermal stability of the cementitious composite were investigated using thermal analysis, and the TG-DTG analyses are presented in Figure 5. The first broad endothermic peak located between 49.8–130 °C is related to the water evaporation. The second endothermic peak located between 400.2–472.5 °C is related to the dehydration of calcium hydroxide. Figure 5a–d exhibit these two endothermic peaks, which indicate the formation of similar hydration products. This suggests that the addition of NS-CNT composite does not affect the hydration reaction. However, the presence of CNTs may affect the hydration kinetics of the cementitious composite due to the total mass loss of 9.8% and 8.92% in control groups 5 and 6, respectively, as shown in Figure 5e. The experimental groups exhibit lower mass losses as compared to the control groups, which suggests that calcium hydroxide is converted into another hydration product [29]. Based on Figure 5a, the total mass loss of G-CNT10.8 + NS7.2 and its mass loss at the second peak are the smallest, at 7.78% and 5.59%, respectively. This suggests more calcium hydroxide is converted into calcium silicate in G-CNT10.8 + NS7.2, which is consistent in its compressive and tensile strength. Based on Figure 5b, total mass loss of OH-G-CNT10.8 + NS7.2 and its mass loss at the second peak are the smallest, at 7.69% and 5.57%, respectively, which indicate that more calcium hydroxide is converted into calcium silicate in OH-G-CNT10.8 + NS7.2. Based on Figure 5c, total mass loss of S-WMNT10.8 + NS7.2 and its mass loss at second peak are the smallest, at 7.46% and 5.17%, respectively, which indicate that more calcium hydroxide is converted to calcium silicate in S-WMNT10.8 + NS7.2. Based on Figure 5d, the total mass loss of OH-S-WMNT10.8 + NS7.2 and its mass loss at the second peak are the smallest, at 7.58% and 5.37%, which suggest more calcium hydroxide is converted to calcium silicate in OH-S-WMNT10.8 + NS7.2.

Figure 6a shows the cumulative heat evolution and heat formation rate (up to 72 h) of the composite cementitious composite paste during hydration for various additives. The first peak is typically related to the wetting of the cementitious composite powder, the dissolution of silicate and gypsum, and the formation of ettringite. After the induction period, the second peak is mainly due to hydration, which results in the formation of C-S-H and calcium hydroxide, and the second peak corresponds to the acceleration zone associated with the formation of sulfate. This indicates that the greater the dispersion of the cementitious composite particles, the better the hydration effect of the cementitious composite can be realized. In most cases, with the addition of NS and CNTs, the NS-CNT composite leads to a short dormancy period, which indicates that the particles with composite structures accelerate the hydration of cementitious composite. These NS-CNT composites can act as nucleation sites due to their large surface areas. The acceleration effect could be identified by the following parameters: (a) The change in the slope of the curve during the acceleration period, (b) the time needed to reach the maximum main peak, and (c) the heat release of the main peak. As shown in Figure 6a, the exothermic energy of the main peak for the composite containing G-CNTs is the highest in the series of pastes modified by the nanoparticles. Different NS-CNT composite materials have different acceleration periods. The different slopes indicate the heat release rate, but the overall slope is similar, indicating that the experimental group containing activated nano-silica-carbon nanotube composite structure has higher heat release than the control group 5 and 6. Moreover, the cumulative heat release of the cementitious composite paste containing activated NS-CNT composite is larger than that of the cementitious composite paste which contained untreated NS-CNT composite. It is also larger than that of control group 5 without any additives. Figure 6b shows that the group with the highest heating rate is the experimental group that contained activated G-CNTs. The next highest heating rate is the other experimental groups that contained CNTs. The result of the control group 6 containing untreated NS-CNTs is between those of the experimental groups and that of the control group 5. This result proves that the activated NS-CNT composite can accelerate the hydration of cementitious composite.

### 3.4. SEM-BET Analysis

SEM and BET measurements were conducted to investigate the morphologies of the composites. The SEM image of the untreated NS/trifluoropropyltrimethoxysilane mixture (Figure 7a) reveals a small amount of NS distributed on the surface of trifluoropropyltrimethoxysilane. Interestingly, the activated NS/trifluoropropyltrimethoxysilane mixture presented in Figure 7b shows an evenly distributed NS on the surface of trifluoropropyltrimethoxysilane. This is because of the dehydration–condensation reaction between the Si-OH (formed after the hydrolysis of trifluoropropyltrimethoxysilane) and the -OH group on the surface of the NS, which results in close binding. The untreated NS-CNT composite shown in Figure 7c shows a small amount of NS adhered to the surfaces of the CNTs. However, after activating NS, NS is able to be uniformly attached onto the surface of CNTs as shown in Figure 7d. The SEM image of control group 5 (Figure 7e) shows the hydration product of cementitious composite, calcium hydroxide, and hydrated calcium silicate. Figure 7f shows that the NS on the surfaces of the CNTs formed irregular ellipsoids of 3–12 μm in length and 2–10 μm in diameter, under the hydrolysis and condensation of trifluoropropyltrimethoxysilane.

Figure 7g shows the structure of the untreated NS-CNTs composite in cementitious composite. A few ellipsoids are formed, and there are only a few pore fillings. Figure 7h shows the structures of the activated NS-CNTs composite in cementitious composite. As the size range of the composite is basically within the pore range of the cementitious composite and the surface of the composite structure contains NS, the Si-OH produced during the hydrolysis of trifluoropropyltrimethoxysilane could also undergo a dehydration–condensation reaction with (CaOH_2_). Therefore, the composite could fill into the pores of the cementitious composite hydration products, and it could combine well with the cementitious composite hydration product.

Figure 8a shows the physical adsorption and desorption curves of the cementitious composite with different compositions. The adsorbed and desorbed amounts of control group 5 are larger than those of control group 6. The adsorbed and desorbed amounts in the experimental group are the smallest, which indicate that the relative porosity of the experimental group is smaller. Figure 8b shows the porosity distribution curves of cementitious composite with different compositions.

The cementitious composite containing NS-CNT composite possesses a low porosity with a pore size distribution in the range of 10–100 nm. The BET results further confirm that NS-CNT composite can reduce the porosity and specific surface area of the cementitious composite after hydration.

### 3.5. Anti-Freeze–Thaw Damage, Electrochemical Corrosion Test, and Contact Angle Test

To verify the influence of NS-CNT composite on the freeze–thaw resistance, durability, and corrosion resistance of the cementitious composite, freeze–thaw tests and electrochemical corrosion experiments were carried out. Figure 9 shows the comparison of freeze–thaw mass loss between the experimental group and the control group in every 30 freeze–thaw cycles. The total mass losses of control groups 5 and 6 after 210 freeze–thaw cycles are 8.35% and 6.87%, respectively, while the mass losses of the experimental groups are maintained between 3.39% and 5.90%. Among the experimental groups, G-CNT3.6 + NS14.4, G-CNT7.2 + NS10.8, and S-WMNT10.8 + NS7.2 show the best freeze–thaw resistances, with total mass losses of 3.56%, 3.61%, and 3.39%, respectively. After 120 freeze–thaw cycles, 0.13–0.26% mass losses are observed for the experimental groups, and there are no mass losses after 210 freeze–thaw cycles for the fourth experimental group. This proves that the NS-CNT composite material, by embedding into the voids of cementitious composite hydration products, reduces the porosity and the natural corrosion resistance of carbon nanotubes, which can enhance the freeze–thaw resistance of cementitious composite.

The corrosion resistance experiments were carried out in NaCl (3.5 wt %) at 26 V for 20 s, with the setup illustrated in Figure 10a. Figure 10d–g shows the photographs of all groups after the experiment. Figure 10b,c present the results of the electrochemical corrosion of control groups 5 and 6, respectively. The contact angles of all experimental groups are marked in the figure. Contact angle measurement is to verify that the nano-silica-carbon nanotube composite structure enhances the waterproofness of cement materials under the combined action of trifluoropropyltrimethylsiloxane. The larger the contact angle, the better the waterproofness. The experimental results show that the surface of the copper rods in the experimental groups with activated NS-CNT composite remains uncorroded, and the contact angles are between 76.5° and 92.5°. A large surface area of the copper rod in control group 5 is corroded, and the contact angle is 41°. The surface of control group 6 is uncorroded, and the contact angle is 71°. This proves that the activated NS-CNT could improve the corrosion resistance of the cementitious composite. Together with trifluoropropyltrimethoxysilane, it also provided water resistance and corrosion resistance for cementitious composite.

## 4. Conclusions

The following conclusions can be drawn based on the results of this study.

NS-CNT elastic composite synthesized by the method described in this paper improved the physical and chemical properties of the cementitious composite materials. The activated NS could accelerate the hydrolysis of trifluoropropyltrimethoxysilane. Trifluoropropyltrimethoxysilane was successfully coated on the surface of the cementitious composite particles. The formation of NS-CNT composite was due to the dehydration–condensation reaction of Si-OH after the hydrolysis of trifluoropropyltrimethoxysilane on the surface with the -OH groups of Ca(OH)_2_, which were interconnected on the surface of the Ca(OH)_2_, filling in the voids. Furthermore, the composite could accelerate the hydration of cementitious composite and promote the further hydration of Ca(OH)_2_. The activated NS-CNT composite accelerated the hydration of cementitious composite, which effectively filled the voids of the cementitious composite hydration products. This reduced the porosities and specific surface areas of the cementitious composite hydration products, which was reflected in the mechanical properties of the materials. The results of the freeze–thaw tests, electrochemical corrosion tests, and contact angle measurements showed that the activated NS-CNT could effectively improve the freeze–thaw and corrosion resistances of the cementitious composite.

## Figures and Tables

**Figure 1 materials-13-03737-f001:**
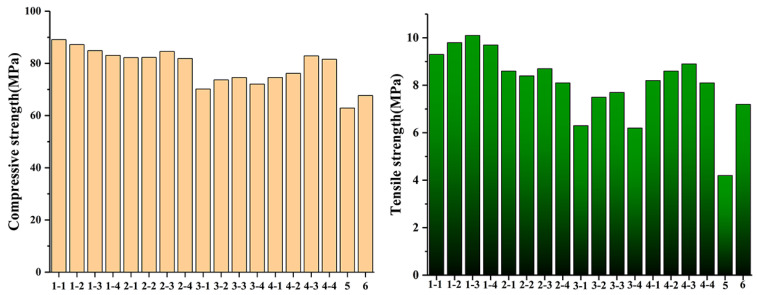
Compressive and tensile strengths of specimens.

**Figure 2 materials-13-03737-f002:**
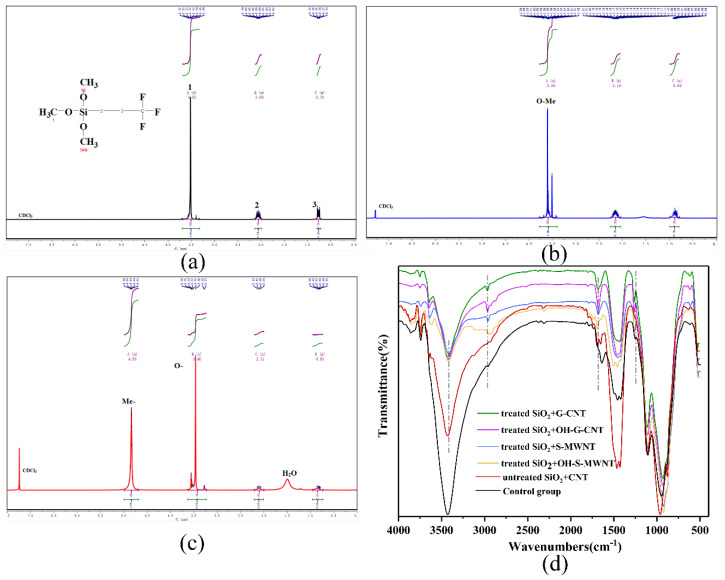
^1^H MR spectra of (**a**) trifluoropropyltrimethoxysilane, (**b**) untreated NS/trifluoropropyltrimethoxysilane, and (**c**) activated NS/trifluoropropyltrimethoxysilane, all tested in deuterated chloroform. (**d**) FTIR spectra of G-CNT10.8 + NS7.2, G-CNT10.8 + NS7.2, S-WMNT10.8 + NS7.2, WMNT10.8 + NS7.2, control group 5, and control group 6.

**Figure 3 materials-13-03737-f003:**
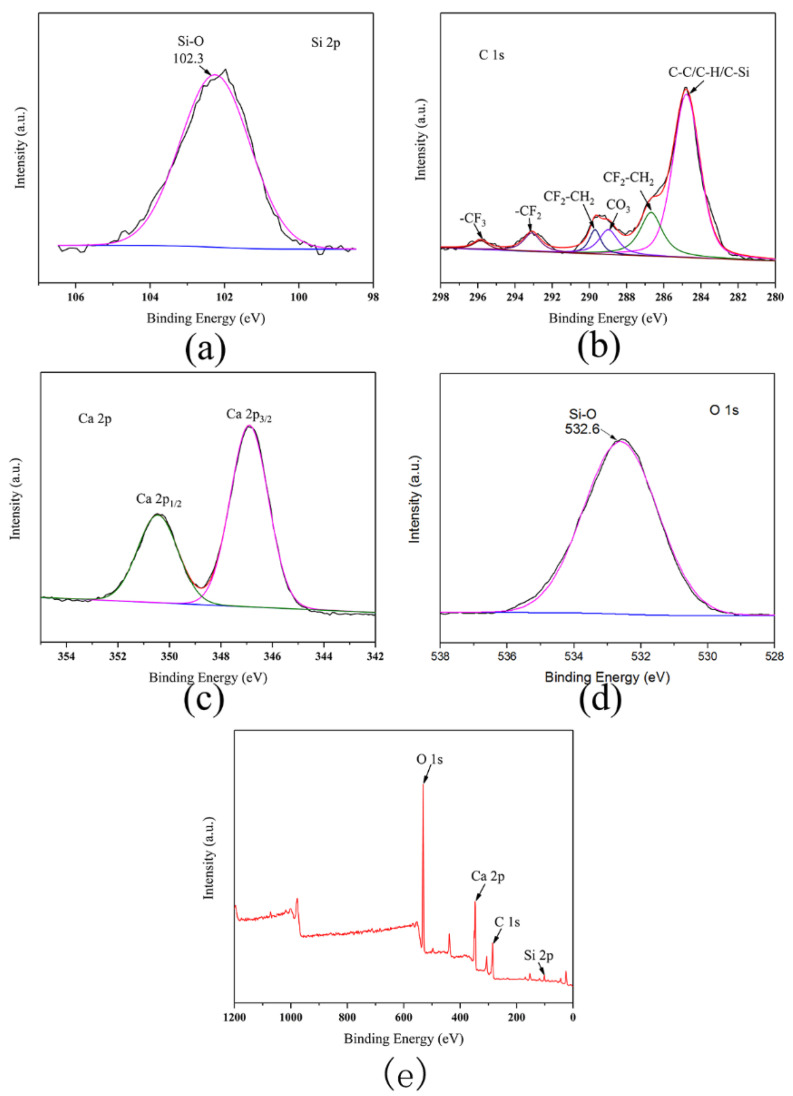
(**a**) Distribution of O, (**b**) C, (**c**) Ca, and (**d**) Si on the surfaces of the cementitious composite containing activated NS-CNTs composite. (**e**) Elemental distributions on the surfaces of cementitious composite containing activated NS-CNTs composite.

**Figure 4 materials-13-03737-f004:**
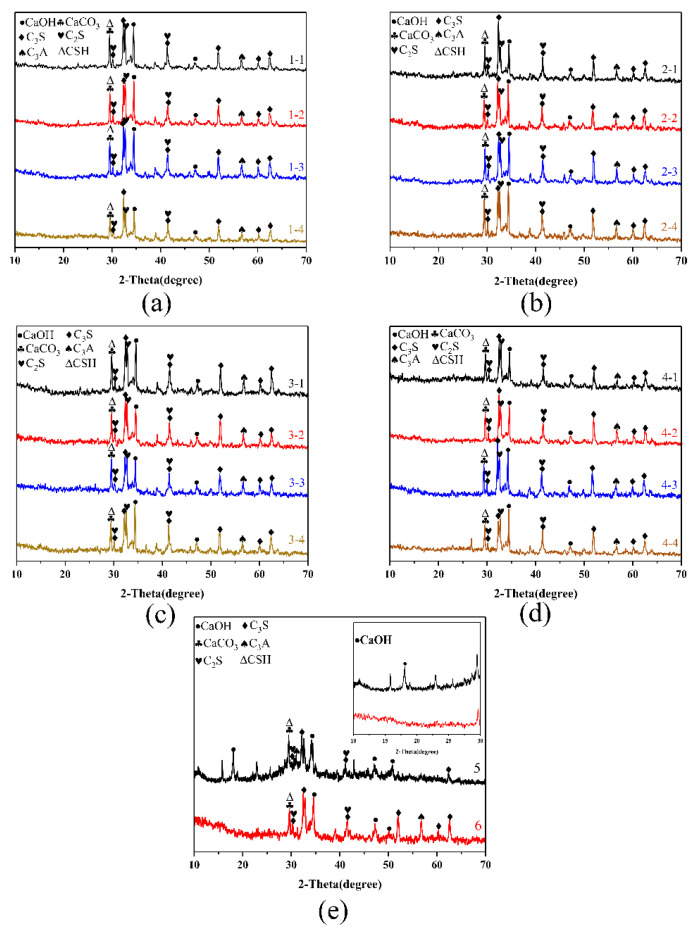
XRD patterns of (**a**) G-CNTs/NS, (**b**) OH-G-CNTs/NS, (**c**) S-MWNTs/NS, (**d**) OH-S-MWNTs/NS, and (**e**) control groups 5 and 6.

**Figure 5 materials-13-03737-f005:**
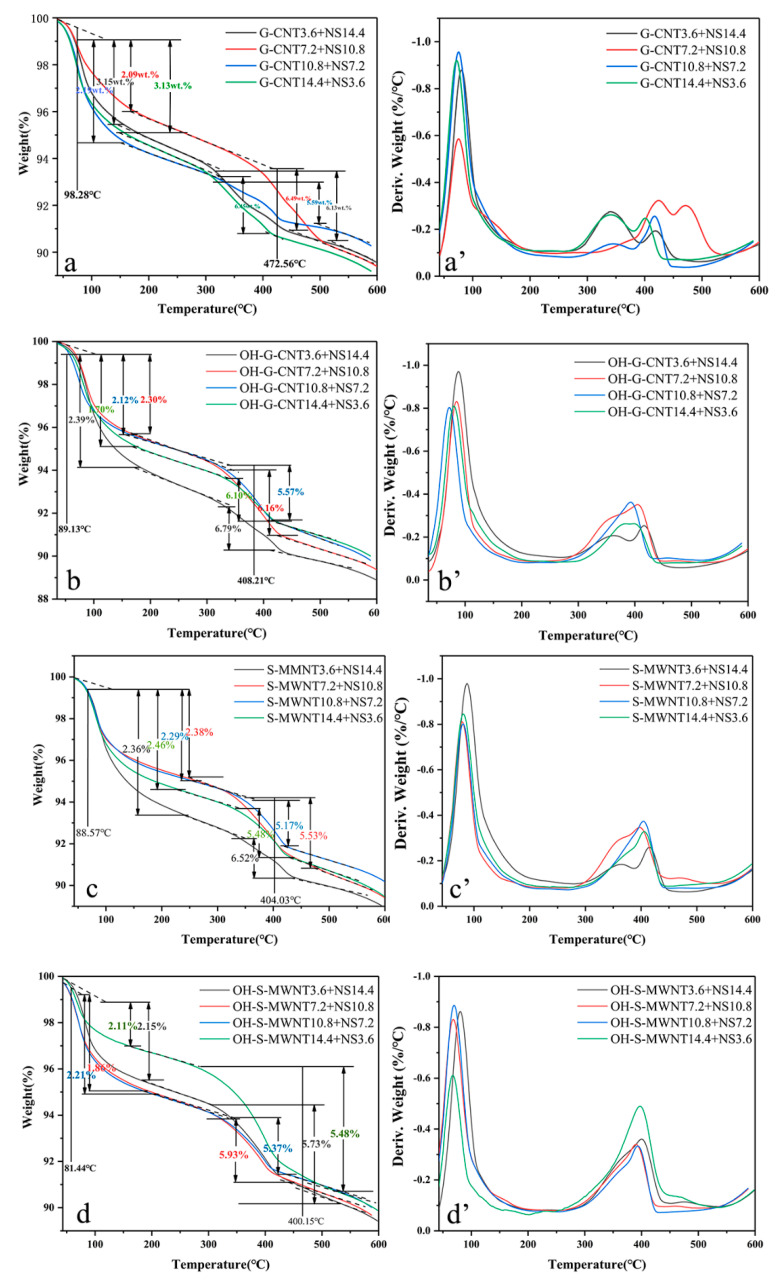
TG and TGA analysis of (**a**,**a’**) G-CNTs/NS, (**b**,**b’**) OH-G-CNTs/NS, (**c**,**c’**) S-MWNTs/NS, (**d**,**d’**) OH-S-MWNTs/NS, and (**e**,**e’**) control groups 5 and 6.

**Figure 6 materials-13-03737-f006:**
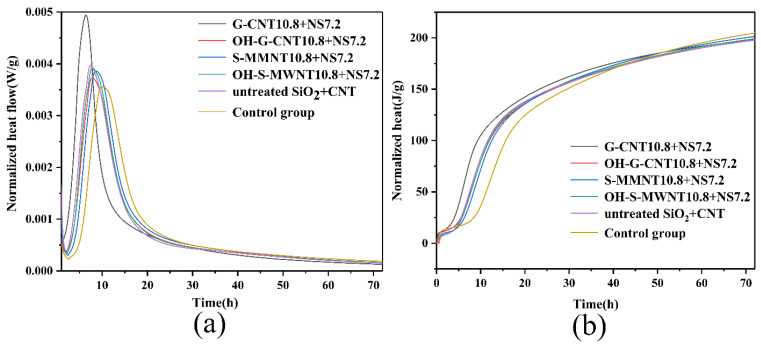
(**a**) Cumulative heat release. (**b**) Heat release rate.

**Figure 7 materials-13-03737-f007:**
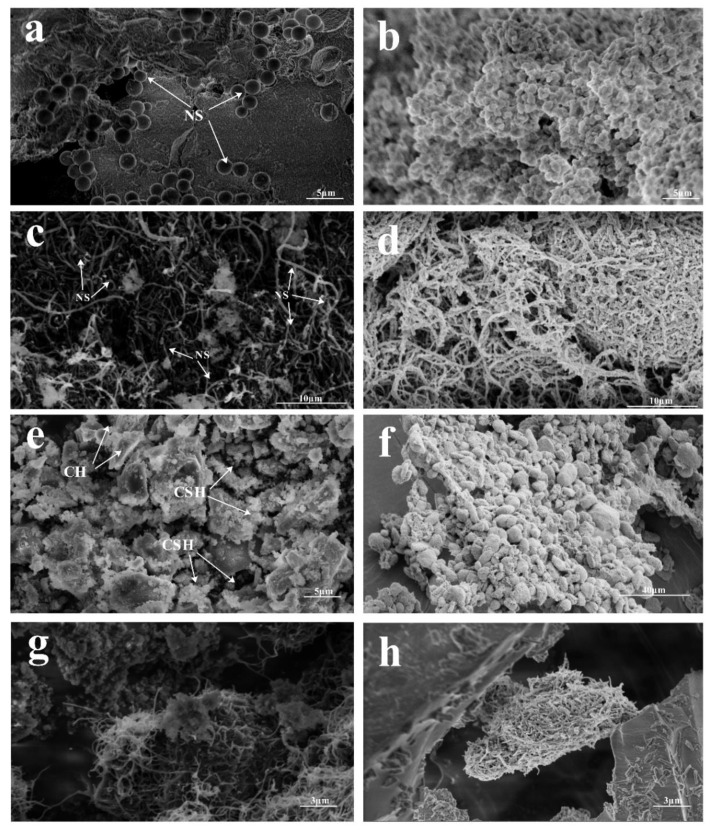
SEM images of (**a**) untreated NS-CNTs composite, (**b**) activated NS-CNTs composite, (**c**) untreated NS-CNTs composite in cementitious composite, (**d**) composite structure of activated NS-CNTs, (**e**) control group 5, (**f**) composite structure of activated NS-CNTs forming ellipsoids, (**g**) morphology of untreated NS-CNTs composite in cementitious composite, and (**h**) morphology of activated NS-CNT composite structure in cementitious composite.

**Figure 8 materials-13-03737-f008:**
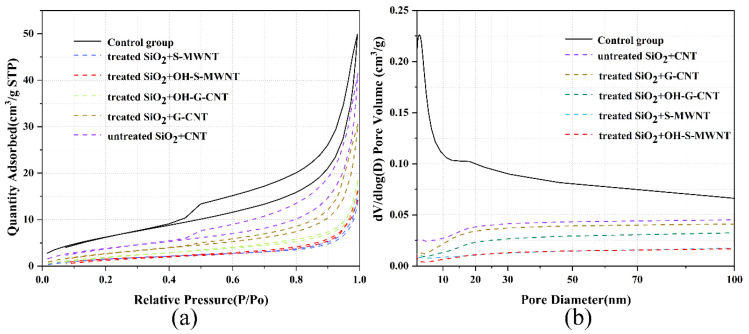
(**a**) Physical adsorption and desorption curves. (**b**) Porosity curves.

**Figure 9 materials-13-03737-f009:**
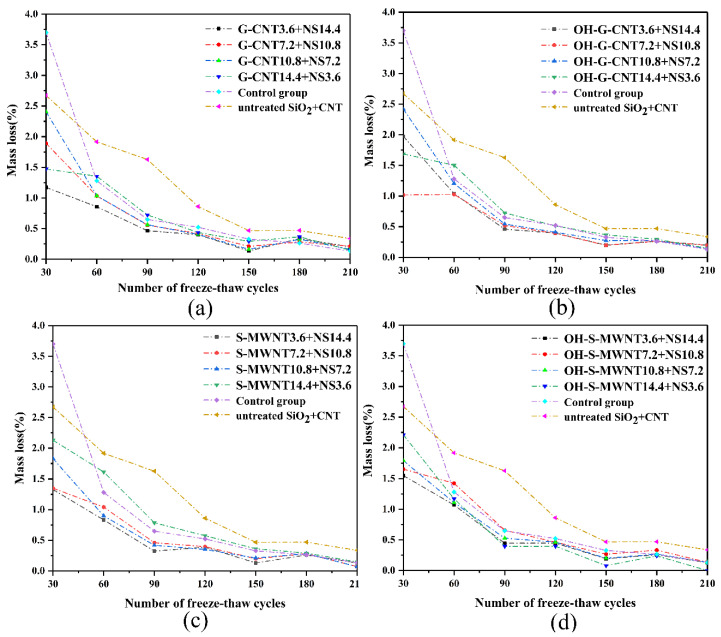
Comparison of freeze–thaw losses between control groups 5 and 6 and experimental groups (**a**) 1, (**b**) 2, (**c**) 3, and (**d**) 4.

**Figure 10 materials-13-03737-f010:**
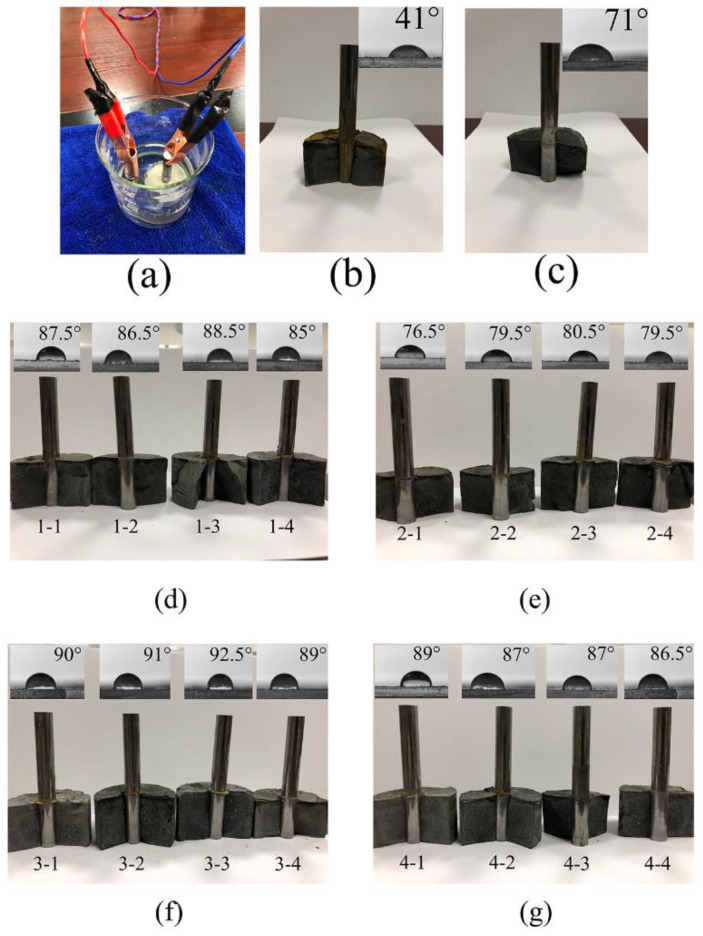
(**a**) Experimental device. (**d**–**g**) Results of corrosion resistance and contact angle after electrochemical corrosion of experimental groups. Results of corrosion resistance and contact angle of (**b**) control group 5, and (**c**) control group 6 after electrochemical corrosion.

**Table 1 materials-13-03737-t001:** Properties of carbon nanotubes.

Type of Carbon Nanotube	Diameter (nm)	Length (μm)	Purity	Specific Surface Area (m^2^/g)
Nano silica	30	—	99.9%	>600
1: G-CNT	8–15	50	99.9%	>233
2: OH-G-CNT	8–15	50	99.9%	>233
3: S-MWNT	10–20	0.5–2	99.9%	>233
4: OH-S-MWNT	8–15	0.5–2	99.9%	>233

G: Graphitization; OH: Hydroxyl; S: short; CNT: Carbon nanotube; MWCNT: multi-wall Carbon nanotube supporting information. G-grade oil-well cementitious composite and ultra-fine Portland cementitious composite were produced by Shandong Kangjing New Material Technology Co., Ltd. (Shandong, China), and their relevant parameters are shown in Table 2.

**Table 2 materials-13-03737-t002:** Oxides and mineralogical phase composition of cementitious composite.

Item	G oil Well Cementitious Composite Mass (%)	Ultrafine Portland Cementitious Composite Mass (%)
SiO_2_	22.10	19.20
Al_2_O_3_	2.90	4.20
Fe_2_O_3_	4.61	2.77
CaO	65.10	62.60
MgO	1.6	1.84
SO_3_	2.2	4.41
K_2_O	0.35	0.91
Na_2_O	0.09	0.15
LOI (loss on ignition)	0.5	3.23
Mineral composition according to Bogue calculation (%)		
C_3_S	64	62.9
C_2_S	13.6	14.2
C_3_A	1	1
C_4_AF	16	16.3

Trifluoropropyltrimethoxysilane (chemical analysis pure reagent) with a concentration of 96% was produced by Macklin Company. Defoaming agent was a mortar defoamer manufactured by Foshan Jingqi Chemical Technology Co., Ltd. (Foshan, China). The water-reducing admixture was a polycarboxylic acid high-performance water-reducing admixture manufactured by Shaanxi Qinfen Building Materials Co., Ltd (Shanxi, China). Tap water was used. The content of each component is shown in Table 3.

**Table 3 materials-13-03737-t003:** Compositions of different mixes and their denotations.

Mixture	Ultrafine Portland Cementitious Composite (kg/m^3^)	Water (kg/m^3^)	Super-Plasticizer	Defoamer	CNT	Nano Silica	SCA
1-1	900	342	9	1.8	3.6	14.4	13.5
1-2	900	342	9	1.8	7.2	10.8	13.5
1-3	900	342	9	1.8	10.8	7.2	13.5
1-4	900	342	9	1.8	14.4	3.6	13.5
2-1	900	342	9	1.8	3.6	14.4	13.5
2-2	900	342	9	1.8	7.2	10.8	13.5
2-3	900	342	9	1.8	10.8	7.2	13.5
2-4	900	342	9	1.8	14.4	3.6	13.5
3-1	900	342	9	1.8	3.6	14.4	13.5
3-2	900	342	9	1.8	7.2	10.8	13.5
3-3	900	342	9	1.8	10.8	7.2	13.5
3-4	900	342	9	1.8	14.4	3.6	13.5
4-1	900	342	9	1.8	3.6	14.4	13.5
4-2	900	342	9	1.8	7.2	10.8	13.5
4-3	900	342	9	1.8	10.8	7.2	13.5
4-4	900	342	9	1.8	14.4	3.6	13.5
5	900	342	9	1.8			
6	900	342	9	1.8	10.8	7.2	13.5

CNT, carbon nanotube; SCA, silane coupling agent.

## Data Availability

The data used to support the findings of this study are included within the article.

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
