# Peer review of "High-Strength, Waterproof, Corrosion-Resistant Nano-Silica Carbon Nanotube Cementitious Composites"

_materials, 2020, doi:10.3390/ma13173737_

Round 1
Reviewer 1 Report
This paper reports some interesting experimental results about cementitious composites based on nano-silica carbon nanotube additions and to some extent, the results elucidate the effects of the groups studied. As such, I consider this paper to be of great interest to future journal readers. Nevertheless, I suggest to the authors small corrections and attention to its writing should be made.
I start with the title, which I suggest changing. Perhaps, “High-strength, waterproof, corrosion-resistant of nano-silica carbon nanotube cementitious composites”. Or ….of cementitious composites with structures based on nano silica carbon nanotube.
Keywords: please start with the most generic term and end with the most specific.
Introduction chapter: in this chapter the problem contextualization is well defined and presented. For example, the reader understands that the question of nanomaterials compactibility and dispersibility in cementitious composites is the main problem to be solved. However, the authors could better remember to readers in the last introduction paragraph, which is the main study pourpose and how you want to achieve.
Some abbreviations must be described in full at the first moment they are presented.
Subchapter 3.1
What are the mechanisms that lead to improved compressive and tensile strengths? Please comment them. How many samples for each group were tested? How about the standard deviations results?
Please correct on lines 211, 234, 235 and others where "cementitious composite composite" is written, as this is a redundancy has no meaning. “Cementitious composite” is enough.
Please review Fig. 2 which is difficult to read
Conclusions:
Furthermore, I only suggest that the conclusions reinforce the fact that the main objective of the study has been achieved.
Author Response
1:The title of the paper has been revised to:High-strength, waterproof, corrosion-resistant of nano-silica carbon nanotube cementitious composites.
2:Keywords: please start with the most generic term and end with the most specific.
I'm very sorry that I can only keep the keywords unchanged, because the keywords given express the important part of this article. Maybe you can give me some better suggestions for modification. Thank you very much.
3:Introduction chapter: in this chapter the problem contextualization is well defined and presented. For example, the reader understands that the question of nanomaterials compactibility and dispersibility in cementitious composites is the main problem to be solved. However, the authors could better remember to readers in the last introduction paragraph, which is the main study pourpose and how you want to achieve.
Modify the last introduction paragraph to:
In this study, the effect of NS and CNT composite structure on the properties of cement materials was systematically studied, including the mechanical properties, water resistance and corrosion resistance of the cement base. Nuclear Magnetic Resonance (NMR) and Fourier Transform In-fared (FTIR) were used to study the dehydration condensation of nano-silica and trifluoropropyltrimethylsiloxane. Scanning electron microscope (SEM) and X-ray Diffraction (XRD), X-ray Photoelectron Spectroscopy (XPS), TG equipment, and calorimetry have studied the influence of nano-silica and CNT composite structure to enhance the microstructure and hydration of cement slurry. The freeze-thaw resistance experiment is used to study the influence of NS and CNTs composite structure on the corrosion resistance of cement materials. The contact angle experiment is used to study the waterproofness of cement materials. This article also introduces the experimental procedures, results analysis and discussion in detail.
4:Some abbreviations must be described in full at the first moment they are presented.
Modify the first occurrence of NS.CNTs in line 14 to:NS(nano-silica), CNTs(carbon nanotube)
5:Subchapter 3.1
What are the mechanisms that lead to improved compressive and tensile strengths? Please comment them. How many samples for each group were tested? How about the standard deviations results?
The improvement of mechanical properties and the detailed experiment process are further elaborated:
The increase in compressive tensile strength is because the activated nano-silica accelerates the hydrolysis of trifluoropropyl trimethylsiloxane, and undergoes a dehydration condensation reaction with it, and then uniformly adheres to the surface of the carbon nanotubes to form An ellipsoid with a length of 3-12 microns and a diameter of 2-10 microns. This material can be evenly distributed in the voids of cement hydration products, greatly reducing the porosity of cement materials. Three sets of experiments were carried out for each set of mechanical experiments, and the compressive and tensile strength diagrams were redrawn and standard error bars were added.
6:Please correct on lines 211, 234, 235 and others where "cementitious composite composite" is written, as this is a redundancy has no meaning. “Cementitious composite” is enough.
Amendments have been made to lines 210 241 243 269 270 277 278 279 288 290 to change "cementitious composite composite" to "cementitious composite".
7:Please review Fig. 2 which is difficult to read
Figure 2 is redrawn, the peak position is emphasized, the explanation information of the figure is deleted, and only the molecular structure of trifluoropropyltrimethylsiloxane and the code of the corresponding peak position are given ( 1.2.3). The resolution of the infrared spectrum has been adjusted higher, in order to further allow reviewers to consult.
8:Conclusions:
Furthermore, I only suggest that the conclusions reinforce the fact that the main objective of the study has been achieved.
The conclusion part is summarized, and only the results and reasons obtained by the experiment are given.
The following conclusions can be drawn based on the results of this study.
NS-CNT elastic composite synthesized by the method described in this paper improved the physical and chemical properties of the Cementitious composite materials.The activated NS could accelerate the hydrolysis of trifluoropropyltrimethoxysilane.Trifluoropropyltrimethoxysilane was successfully coated on the surface of the Cementitious composite particles.The formation of NS-CNT composite was due to the dehydration–condensation reaction of Si-OH after the hydrolysis of trifluoropropyltrimethoxysilane on the surface with the -OH groups of CaOH, which were interconnected on the surface of the Ca(OH)2, filling in the voids. Furthermore, the composite could accelerate the hydration of Cementitious composite and promote the further hydration of Ca(OH)2.The activated NS-CNT composite accelerated the hydration of Cementitious composite, which effectively filled the voids of the Cementitious composite hydration products. This reduced the porosities and specific surface areas of the Cementitious composite hydration products, which was reflected in the mechanical properties of the materials.The results of the freeze–thaw tests, electrochemical corrosion tests, and contact angle measurements showed that the activated NS-CNT could effectively improve the freeze–thaw and corrosion resistances of the Cementitious composite.

Reviewer 2 Report
TITLE
What does it mean „Cementitious compositeitious”? ...composite, I think?
ABSTRACT
The abstract structure is proper. However, there some grammar errors, unclear and badly constructed sentences, phrases, and even words.
Some grammar errors: Which these products; The high-water resistances of the Cementitious composite-based composites and good durability and corrosion resistance.
A lack of subscript: Ca(OH)2
Why in the full text the cementitious composite is written with a capital letter, as Cementitious? I think it is wrong.
“Cementitious composite-based composites”. Formally, it is proper, but the whole phrase sounds strange.
“further hydration of the Cementitious composite hydration product (CaOH)2 as well as further dehydration–condensation of the surface hydroxyl groups with the inherent hydroxyl groups of (CaOH)2.” I cannot understand this sentence: hydration of hydration product? Calcium hydroxide is a product and not a component? Hydroxyl groups (and not molecules) can be dehydrated and condensed?
INTRODUCTION
The introduction is sufficient to see the problem and state-of-the-art. However, the last part should be limited to the justification of the research and showing its aim. Here, the summary of the results is unnecessarily given. For that and our reasons shown below, the introduction must be modified.
At first, the literature references should be numbered with an increasing order, and here appear, successively, 4, 1, 2, 6-12, 13, 14, 15, 17-19, 20, 21, 22, 5, 23, 12, 24, 25, 30, 31, 32, 33, 34, 35. There is then a lack of cited references: 3, 16, 26-29.
Grammar errors: harfs.With (no space); With the extraction of oil, Cl- erosion can degrade…(rather: at the extraction, when performing the extraction…); have studied the incorporation of nanoparticles enhanced (that enhanced). Also: Majid R Ayatollahi also studied the effect of carbon
nanotubes and nanosilica on the tribological properties of carbon fiber cloth composite materials and
concluded that which indicates enhanCementitious composite in bonding strength between carbon
fiber and epoxy matrix due to the interfacial reinforcing action of the nano-particles[35].
A lack of subscript: SiO2.
The authors give full names and surnames when citing references, for example, “Reza Moghimi Monfared, Majid R.”. Formally, it is not a mistake, but the usually applied writing style is different such as “Monfrared et al.”.
EP resin: for the first time give always the full name and no abbreviation (epoxy resin (EP)). The same remark for GO, MWCNT-81 G/VMQ
“strengthens the fiber-matrix interface strength, and then strengthens the surrounding matrix to increase the strength”; better “increases the fiber…”, but the second part of this sentence is unclear.
MATERIALS AND METHODS
The section is well-constructed. However, among analytical techniques, the corrosion tests and mechanical tests should be described here and not in the Results section. There is no description of a model and a manufacturer with a location for several techniques, please check.
Some grammar errors: 1H Nuclear Magnetic Resonance (NMR) with a prepolymer of PSR were performed; According to the China National Standard JTG E30-184 2005,The Methods of Cementitious composite and Cementitious composite for Highway Engineering;
RESULTS AND DISCUSSION
Mechanical properties analysis
Why the Tables are called S1, S2, S3, and not Tables 1, 2, 3?
“As shown in Fig. 1, the measured compressive and tensile strengths are 102.04% and 140.47% higher
than those of the control group, respectively, and 53.51% and 71.43% higher than those of the group
without hydrothermal activation treatment”. It is not true. I presume that the measured strengths constitute 102.04% … and are higher of 102.4%. Besides, please give also the ratio of corresponding values (new cement to the reference cement) in a new table, it is also necessary for further discussion of determinants.
NMR, FTIR, and XPS analysis
There is Fig. 2d described in the text.
There is no Fig. 3f Should be 3e?).
XRD, TG, and isothermal calorimetry analysis
There is no c, c` inside the appropriate pictures in Fig. 5.
“but the overall slopes are similar, which is higher than those of control groups”; unclear.
There is no description of Fig. 7c in the text.
“(h) morphology of untreated NS-CNTs composite in Cementitious composite, and (i) morphology of
activated NS-CNT composite structure in Cementitious composite.”. Change (h) to (g) and (i) to (h) letters.
Anti-freeze–thaw damage, electrochemical corrosion test, and contact angle test
There is no so far any mention about measurements of contact angle (it should be in the experimental part). Please explain to the reader what is the relation between wettability and corrosion resistance.
All critiques points are in color in the attached and reviewed manuscript.

Author Response
1:TITLE
What does it mean „Cementitious compositeitious”? ...composite, I think?
The title of the paper has been revised to:High-strength, waterproof, corrosion-resistant of nano-silica carbon nanotube cementitious composites.
2: ABSTRACT
The abstract structure is proper. However, there some grammar errors, unclear and badly constructed sentences, phrases, and even words.
Some grammar errors: Which these products; The high-water resistances of the Cementitious composite-based composites and good durability and corrosion resistance.
These products represent Polymerization product of nano-silica and trifluoropropyl trimethylsiloxane. The high-water resistances of the cementitious composite-based composites and good durability and corrosion resistance. change into The cementitious composite containing the activated NS-CNT exhibit high mechanical strengths,the high water resistances and good durability and corrosion resistance.
A lack of subscript: Ca(OH)2
Ca(OH)2 has been rewritten as Ca(OH)2.
Why in the full text the cementitious composite is written with a capital letter, as Cementitious? I think it is wrong.
Cementitious in the full text has been rewritten as cementitious.
“Cementitious composite-based composites”. Formally, it is proper, but the whole phrase sounds strange.
Cementitious composite-based composites were modified to cementitious composite.
“further hydration of the Cementitious composite hydration product (CaOH)2 as well as further dehydration–condensation of the surface hydroxyl groups with the inherent hydroxyl groups of (CaOH)2.” I cannot understand this sentence: hydration of hydration product? Calcium hydroxide is a product and not a component? Hydroxyl groups (and not molecules) can be dehydrated and condensed?
The activated NS-CNT composite material effectively promotes the further hydration of (CaOH)2 in the cement to form hydrated calcium silicate, and further dehydration-condensation between the surface hydroxyl group of the composite material and the inherent hydroxyl group of (CaOH)2.
This study aims to prepare a nano-silica-carbon nanotube (NS-CNT) elastic composite using NS(nano-silica), CNTs(carbon nanotube), and (D3F)trifluoropropyltrimethoxysilane. The results show that the activated NS could promote the hydrolysis of D3F. Polymerization product of nano-silica and D3F are uniformly adhered onto the surfaces of CNTs, thereby forming a NS-CNT composite. The composite is composed of irregular ellipsoids with 3–12 μm in length and 2–10 μm in diameter. The activated NS-CNT composite material effectively promotes the further hydration of (CaOH)2 in the cement to form hydrated calcium silicate, and further dehydration-condensation between the surface hydroxyl group of the composite material and the inherent hydroxyl group of (CaOH)2. The cementitious composite-based composites containing the activated NS-CNT exhibit high mechanical strengths,the high water resistances and good durability and corrosion resistance. The chemical characterizations reveal the morphology, nucleation mode of the composite, and its influence on the hydration structure and products of cementitious composite.
3: INTRODUCTION
The introduction is sufficient to see the problem and state-of-the-art. However, the last part should be limited to the justification of the research and showing its aim. Here, the summary of the results is unnecessarily given. For that and our reasons shown below, the introduction must be modified.
Modify the last introduction paragraph to:
In this study, the effect of NS and CNT composite structure on the properties of cement materials was systematically studied, including the mechanical properties, water resistance and corrosion resistance of the cement base. Nuclear Magnetic Resonance (NMR) and Fourier Transform In-fared (FTIR) were used to study the dehydration condensation of nano-silica and trifluoropropyltrimethylsiloxane. Scanning electron microscope (SEM) and X-ray Diffraction (XRD), X-ray Photoelectron Spectroscopy (XPS), TG equipment, and calorimetry have studied the influence of nano-silica and CNT composite structure to enhance the microstructure and hydration of cement slurry. The freeze-thaw resistance experiment is used to study the influence of NS and CNTs composite structure on the corrosion resistance of cement materials. The contact angle experiment is used to study the waterproofness of cement materials. This article also introduces the experimental procedures, results analysis and discussion in detail.
At first, the literature references should be numbered with an increasing order, and here appear, successively, 4, 1, 2, 6-12, 13, 14, 15, 17-19, 20, 21, 22, 5, 23, 12, 24, 25, 30, 31, 32, 33, 34, 35. There is then a lack of cited references: 3, 16, 26-29.
Renumbered references
Grammar errors: harfs.With (no space);
Space added.
With the extraction of oil, Cl- erosion can degrade…(rather: at the extraction, when performing the extraction…); have studied the incorporation of nanoparticles enhanced (that enhanced). Also: Majid R Ayatollahi also studied the effect of carbon
With the extraction of oil, Cl- erosion can degrade the rebar performance and shorten the service life of the reinforced Cementitious composite [4]. Rewritten as
At the extraction of oil.Cl-erosion will corrode steel and shorten the service life of cement-based composite materials [4].
A lack of subscript: SiO2.
Added subscript: SiO2 in lower 75 lines.
The authors give full names and surnames when citing references, for example, “.”. Formally, it is not a mistake, but the usually applied writing style is different such as “Monfrared et al.”.
Reza Moghimi Monfared, Majid R. Rewritten as Monfared et al.
EP resin: for the first time give always the full name and no abbreviation (epoxy resin (EP)). The same remark for GO, MWCNT-81 G/VMQ
EP is rewritten as epoxy resin (EP); GO is rewritten as MWCNT-81 G/VMQ(GO)
“strengthens the fiber-matrix interface strength, and then strengthens the surrounding matrix to increase the strength”; better “increases the fiber…”, but the second part of this sentence is unclear.
MR Ayatollahi also studied that the combination of carbon nanotubes and carbon nanotubes strengthens the fiber-matrix interface strength, stiffness of the multi-scale composite material[32].
MATERIALS AND METHODS
The section is well-constructed. However, among analytical techniques, the corrosion tests and mechanical tests should be described here and not in the Results section. There is no description of a model and a manufacturer with a location for several techniques, please check.
The mechanical performance test method is added to the MATERIALS AND METHODS section, and simple equipment and test methods for corrosion resistance testing are provided.
Flexural and compressive tests were carried out using a TYE-300D Cementitious composite mortar flexural and compressive test machine (Wuxi Jianyi Instrument Machinery Co., Ltd.). The corrosion resistance test adopts a self-designed device, which contains two electrodes and a power plug. The two electrodes are respectively connected to the copper rod in the cement block sample, and the other end is connected to the copper rod and placed in salt water, and 220V AC is connected.. The corrosion resistance experiments were carried out in NaCl (3.5 wt%) at 26 V for 20 s.
Some grammar errors: 1H Nuclear Magnetic Resonance (NMR) with a prepolymer of PSR were performed; According to the China National Standard JTG E30-184 2005,The Methods of Cementitious composite and Cementitious composite for Highway Engineering;
Change into:Use AVANCE III HD 400 Bruker spectrometer (Switzerland Bruker) for 1H nuclear magnetic resonance (NMR) test, The test standard adopts the Chinese national standard JTG E30-2005.
RESULTS AND DISCUSSION
Mechanical properties analysis
Why the Tables are called S1, S2, S3, and not Tables 1, 2, 3?
Rename the table to Table1,2,3
As shown in Fig. 1, the measured compressive and tensile strengths are 102.04% and 140.47% higher than those of the control group, respectively, and 53.51% and 71.43% higher than those of the group without hydrothermal activation treatment”. It is not true. I presume that the measured strengths constitute 102.04% … and are higher of 102.4%. Besides, please give also the ratio of corresponding values (new cement to the reference cement) in a new table, it is also necessary for further discussion of determinants.
Change into:
As shown in Fig. 1, the measured compressive and tensile strengths are 41.65% and 140.47% higher than those of the control group, respectively, and 31.6% and 40.3% higher than those of the group without hydrothermal activation treatment.
My calculation method is to define the compressive strength and tensile strength of the control group as A1, B1; the compressive strength and tensile strength of the experimental group with activated nano-silica-carbon nanotube material are A2, B2; The compressive strength and tensile strength of the experimental group of unactivated nano-silica-carbon nanotube materials are A3, B3; the increase in strength is (A2-A1)/A1; (A2-A3)/A3; (B2-B1)/B1; (B2-B3)/B3;
NMR, FTIR, and XPS analysis
There is Fig. 2d described in the text.
There is no Fig. 3f Should be 3e?).
Figure 2d is used to describe the FTIR
Fig 3f should be 3e. Has been modified
XRD, TG, and isothermal calorimetry analysis
There is no c, c` inside the appropriate pictures in Fig. 5.
Figure 5c.c’ is explained and described in the lower line 330.331
“but the overall slopes are similar, which is higher than those of control groups”; unclear.There is no description of Fig. 7c in the text.
Different NS-CNT composite materials have different acceleration periods. The different slopes indicate the heat release rate, but the overall slope is similar, indicating that the experimental group containing activated nano-silica-carbon nanotube composite structure has higher heat release than the control group5 And 6.
Fig 7c is corrected again on line 386.
“(h) morphology of untreated NS-CNTs composite in Cementitious composite, and (i) morphology of activated NS-CNT composite structure in Cementitious composite.”. Change (h) to (g) and (i) to (h) letters.
Change (h) to (g) and (i) to (h) letters.
Anti-freeze–thaw damage, electrochemical corrosion test, and contact angle test
There is no so far any mention about measurements of contact angle (it should be in the experimental part). Please explain to the reader what is the relation between wettability and corrosion resistance.
The contact angle measurement uses the OCA series video optical contact angle measuring instrument developed and produced by the German Dataphysics company. Contact angle measurement range: 0~180°; measurement accuracy: ±0.1°. Contact angle measurement is to verify that the nano-silica-carbon nanotube composite structure enhances the waterproofness of cement materials under the combined action of trifluoropropyltrimethylsiloxane. The larger the contact angle, the better the waterproofness.
All critiques points are in color in the attached and reviewed manuscript.
